# ATRX, DAXX or MEN1 mutant pancreatic neuroendocrine tumors are a distinct alpha-cell signature subgroup

Chang S. Chan[1,2], Saurabh V. Laddha [1], Peter W. Lewis [3], Matthew S. Koletsky[4], Kenneth Robzyk[5], Edaise Da Silva[5], Paula J. Torres[5], Brian R. Untch[6], Janet Li[6], Promita Bose[7], Timothy A. Chan [7], David S. Klimstra[5], C. David Allis[4] & Laura H. Tang[5]

The commonly mutated genes in pancreatic neuroendocrine tumors (PanNETs) are *ATRX*, *DAXX*, and *MEN1*. We genotyped 64 PanNETs and found 58% carry *ATRX*, *DAXX*, and *MEN1* mutations (A-D-M mutant PanNETs) and this correlates with a worse clinical outcome than tumors carrying the wild-type alleles of all three genes (A-D-M WT PanNETs). We performed RNA sequencing and DNA-methylation analysis to reveal two distinct subgroups with one consisting entirely of A-D-M mutant PanNETs. Two genes differentiating A-D-M mutant from A-D-M WT PanNETs were high *ARX* and low *PDX1* gene expression with *PDX1* promoter hyper-methylation in the A-D-M mutant PanNETs. Moreover, A-D-M mutant PanNETs had a gene expression signature related to that of alpha-cells (FDR *q*-value < 0.009) of pancreatic islets including increased expression of HNF1A and its transcriptional target genes. This gene expression profile suggests that A-D-M mutant PanNETs originate from or transdifferentiate into a distinct cell type similar to alpha cells.

[1] Rutgers Cancer Institute of New Jersey, New Brunswick, NJ 08903, USA. [2] Rutgers Robert Wood Johnson Medical School, New Brunswick, NJ 08901, USA. [3] Epigenetics Theme, Wisconsin Institute for Discovery, University of Wisconsin, Madison, WI, USA. [4] Laboratory of Chromatin Biology and Epigenetics, The Rockefeller University, New York, NY 10065, USA. [5] Department of Pathology, Memorial Sloan-Kettering Cancer Center, New York, NY 10065, USA. [6] Department of Surgery, Memorial Sloan Kettering Cancer Center, New York, NY 10065, USA. [7] Department of Radiation Oncology, Memorial Sloan Kettering Cancer Center, New York, NY 10065, USA. These authors contributed equally: Chang S. Chan, Saurabh V. Laddha. Correspondence and requests for materials should be addressed to C.S.C. (email: chanc3@cinj.rutgers.edu) or to L.H.T. (email: tangl@mskcc.org)

Pancreatic neuroendocrine tumors (PanNETs) or islet cell tumors are a relatively rare neuroendocrine malignancy with an annual incidence of <1 per 100,000 per year[1] (about 1000 new cases per year in the United States) but currently represent the second most common epithelial neoplasm after ductal adenocarcinoma of the pancreas and account for 1–2% of pancreatic tumors. PanNETs were erroneously considered a benign group of neoplasm because they were initially mostly comprised of benign symptomatic insulin-producing tumors (insulinomas). However, in the past three decades, it has become apparent that at least half of all PanNETs are nonfunctional, and they are a heterogeneous group of tumors with often unpredictable and varying degrees of malignancy. As many as 50–80% of PanNETs are associated with synchronous or metachronous metastatic disease[2]. Knowledge of functional PanNETs has evolved from insulinoma to almost a dozen other diverse hormone-secreting tumors. These individual lesions may have specific clinical, pathologic, and genetic associations, including multiple endocrine neoplasia type 1 (MEN-1), tuberous sclerosis, and von Hippel-Lindau (VHL) syndromes. Thus, the entity of PanNET represents a diverse group of heterogeneous neoplasms where combined clinical and pathologic assessment is required to further identify their genetic basis for neoplasia and to define their specific clinical behavior. The nonfunctional tumors require further elucidation to characterize their diverse pathogenesis and to predict outcome with potential biomarkers and molecular signatures. Current classification scheme for PanNETs include grade and stage[1]. The World Health Organization (WHO) classification, which assesses the proliferative index of neoplastic cells, divides PanNETs into low grade (G1), intermediate grade (G2), and high grade (G3). The higher grade PanNETs are generally associated clinically with more aggressive behavior[3]. Poorly differentiated neuroendocrine tumor of the pancreas is extremely rare and clinically aggressive, which represents a different pathogenesis from the well differentiated counterpart[4]. While well-differentiated PanNETs can be successfully treated with surgery, there are few treatments for metastatic PanNETs, and they do not respond to conventional chemotherapy. A greater understanding of PanNET pathogenesis may guide the development of novel therapeutic options.

Molecular studies have identified mutations in MEN1, ATRX, and DAXX to be the most commonly found in PanNETs[5,6] (found in approximately 40, 10, and 20% of tumors, respectively). All three genes play a role in chromatin remodeling. MEN1 is a component of a histone methyltransferase complex[7] that specifically methylate Lysine 4 of histone H3 and functions as a transcriptional regulator. ATRX and DAXX interact to deposit histone H3.3-containing nucleosomes in centromeric and telomeric regions of the genome[8]. Additional mutations in mTOR pathway genes including TSC2, PTEN, and PIK3CA are found in one in six well-differentiated PanNETs[5]. Other reported rare mutations in PanNETs include DNA damage repair genes (MUTYH, CHEK2, BRCA2) and chromatin remodeling gene SETD2[6].

The neuroendocrine cells in the pancreas include alpha, beta, delta, pancreatic polypeptide (pp)-producing and vasoactive intestinal peptide (VIP)-producing cells. The cell of origin for nonfunctional PanNETs is not well established. Here, we genotyped 64 well differentiated PanNETs for mutations in ATRX, DAXX, and MEN1 and performed RNA sequencing ($n = 33$) and DNA methylation ($n = 32$) analysis to identify distinct molecular phenotypes of A-D-M mutant PanNETs which potentially reveals their distinct cell of origin or transdifferentiated state.

## Results

**Clinical annotation and genotyping for _ATRX_, _DAXX_, and _MEN1_**. We initially performed Sanger sequencing to genotype the ATRX, DAXX, and MEN1 genes in 64 individual PanNETs. All cases were histologically confirmed to be well-differentiated PanNETs of WHO G1/G2 grade, and cases of poorly differentiated neuroendocrine carcinoma were excluded. The mean patient age was $52 \pm 1.5$ years (mean ± standard error, ranging from 26–73) with a 59% male population. The locations of the tumors were 38% proximal/mid body, and 62% distal pancreas. Eighty-one percent of the cases were clinically non-functional and the remaining cases included insulinomas, glucagonomas, gastrinomas, and VIPomas. The median size of tumor was $3.6 \pm 0.4$ cm (ranging from 1.0–14.5 cm). Sixty-eight percent of patients had localized disease without distant metastasis at the time of initial diagnosis (Supplementary data 1).

An A-D-M mutant genotype was identified in 58% (37/64) of cases with ATRX, DAXX, MEN1, MEN1/ATRX, and MEN1/DAXX mutations in 8, 16, 20, 3, and 11% cases, respectively (Fig. 1a). The majority of mutations in ATRX, DAXX, and MEN1 were truncation mutations (stopgain or frameshift) and loss of function consistent with their role as tumor suppressors (Supplementary Data 2). Similar to the observations in our previously published data[9], the 5-year disease specific survival was associated with tumor stage ($p$-value < 0.04, log-rank tests), tumor grade (G1 vs G2 $p$-value < 0.02, log-rank tests), and distant metastasis ($p$-value < 0.002, log-rank tests), respectively. Among 44 patients who initially presented with localized disease without distant metastasis, those with the A-D-M mutant genotype had a worse recurrence free survival than those of A-D-M WT genotype (Fig. 1b). Furthermore, in comparison to A-D-M WT PanNETs, the A-D-M mutant PanNETs were associated with larger tumor size ($3.6 \pm 0.6$ cm vs. $5.6 \pm 0.7$ cm, $p$-value < 0.03, log-rank tests) and higher tumor stage (T1 and T2 vs. T3, $p$-value < 0.04, log-rank tests). Other demographic and clinical characteristics (including gender, age, tumor functionality, and lymph node metastasis) revealed no statistically significant differences between the two genotypes of PanNETs.

**Gene expression and DNA methylation reveal two subtypes of PanNETs**. We performed RNA sequencing on 33 randomly selected tumors (19 A-D-M mutant, and 14 A-D-M WT). Unsupervised hierarchical clustering of the top 3000 variable genes across the PanNETs revealed two distinct clusters where almost all A-D-M mutant PanNETs were found in one cluster (Fig. 2a). The grouping of A-D-M mutant PanNETs into one distinct cluster by gene expression was robust to the number of most variable genes used for clustering (Supplementary Figure 1). Principal component analysis (PCA) separated the A-D-M mutant PanNETs from the A-D-M WT PanNETs along the first principal component (corresponding to the component comprising the largest variation in gene expression) (Fig. 2b). The separation of A-D-M mutant PanNETs from A-D-M WT PanNETs by PCA was robust to the number of top variable genes used (Supplementary Figure 2). These data show that A-D-M mutant tumors have a distinct gene expression pattern from that of A-D-M WT PanNETs. Neither hierarchical clustering nor PCA from gene expression revealed further subgrouping of the tumors with single mutations in ATRX, DAXX, or MEN1 or double mutations in ATRX/MEN1 or DAXX/MEN1.

In hierarchical clustering, the A-D-M mutant PanNETs formed a tighter cluster than the A-D-M WT PanNETs. In PCA, the A-D-M mutant PanNETs had smaller variance along PC1 than A-D-M WT PanNETs. Pair-wise correlation of gene expression between all PanNETs, showed a higher correlation among A-D-M mutant

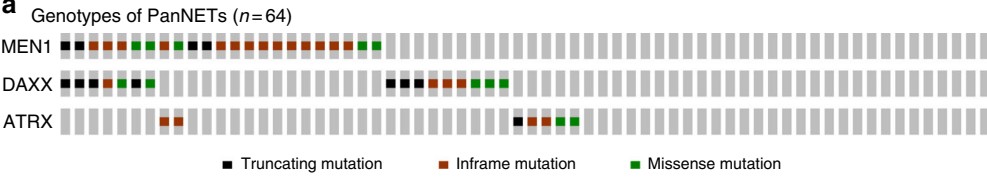

**a** Genotypes of PanNETs (*n* = 64)

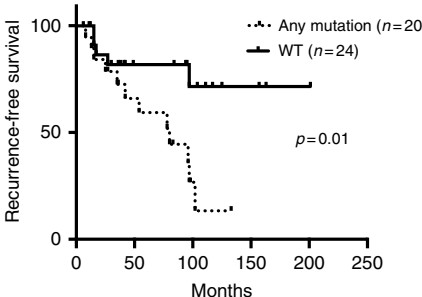

**Fig. 1** Mutational landscape of *ATRX*, *DAXX*, and *MEN1* in PanNETs. **a** Oncoprint mutational profile for PanNETs samples. *ATRX/DAXX/MEN1* mutations were identified in 37/64 (58 %) of PanNETs using Sanger sequencing. **b** Among 44 patients who initially presented with localized PanNETs (without distant metastasis), those with A-D-M mutant genotype had a worse recurrence free survival outcome than those A-D-M WT genotype in their primary tumors. A-D-M mutated samples are annotated as any mutation (*n* = 20) and A-D-M WT samples annotated as WT (*n* = 24)

PanNETs as compared to A-D-M WT PanNETs (Fig. 2c). Among A-D-M mutant PanNETs, mutants with the same genotype (mutations in *ATRX/DAXX/MEN1*) did not show greater gene expression correlation. These data suggest that A-D-M mutant PanNETs are a more homogeneous group compared to A-D-M WT PanNETs.

Within A-D-M mutant or A-D-M WT PanNETs groups, unsupervised clustering and PCA did not reveal differences between primary and metastatic tumors. Top 100 genes with highest variance across all samples separates mutant from A-D-M WT PanNETs and showed relatively high expression of "liver-specific" genes (*APOH, ALDH1A1, FGB, APOC3* etc.) as well as complement and coagulation pathway genes (*SERPINA1, FGA, F10, CP, MT3* etc.) in A-D-M mutant PanNETs (Fig. 2d; Supplementary Figure 3), both in primary (collected in absence of liver tissue) and metastatic tumors. Moreover, the pathological estimate of tumor purity was over 80% for all samples of PanNETs consistent with inference from ESTIMATE[10] (median tumor purity of 90%, Supplementary data 3) showing high tumor purity characteristic of well differentiated PanNETs. In addition, seven A-D-M mutants and one A-D-M WT PanNETs were from the tissue of liver metastases and they had gene expression profile most similar to the genotype group of their primary PanNET counterpart (Fig. 2d). We confirmed the distinct gene expression signature of A-D-M mutant PanNETs in a larger tumor set (47 PanNETs including the 33 PanNETs where RNA sequencing was performed) using gene expression microarray technology. The 14 additional samples are comprised of 3 A-D-M WT PanNETs and 11 A-D-M mutant PanNETs (7 *MEN1* mutant, 2 *DAXX* mutant, and 2 *DAXX/MEN1* mutant)(Supplementary Figure 4).

To investigate epigenetic differences between PanNETs, we used the Illumina 450 K chip to assay the DNA methylation at 411,549 CpG sites in 32 PanNETs. Unsupervised hierarchical clustering of the top 2000 variable DNA methylation sites across the PanNETs revealed two distinct clusters where almost all A-D-M mutant PanNETs were found in one cluster (Fig. 3a). Principal component analysis (PCA) separated the A-D-M mutant PanNETs from the A-D-M WT PanNETs along the first principal component (corresponding to the component comprising the largest variation in DNA methylation) (Fig. 3b). The separation of

A-D-M mutant PanNETs from A-D-M WT PanNETs by PCA was robust to the number of top variable DNA methylation sites used (Supplementary Figure 5). These data reveal that A-D-M mutants PanNETs have a distinct DNA methylation pattern from that of A-D-M WT PanNETs. Neither hierarchical clustering nor PCA revealed differences in DNA methylation sites between the different combinations of genes mutated among the A-D-M mutant PanNETs. Within A-D-M mutant or A-D-M WT PanNETs groups, unsupervised clustering and PCA of DNA methylation did not reveal differences between primary and metastatic tumors.

To investigate the global histone methylation level in PanNETs with and without A-D-M mutations, we performed immunohistochemistry on H3K4me3, H3K9me3, H3K27me3, and H3K36me3 on 36 PanNETs. There was a general trend of lower histone methylation level for *MEN1* mutated PanNETs when compared to WT PanNETs (Supplementary Figure 6 and Supplementary Table 1).

**A-D-M mutant PanNET gene expression resembles that of alpha cells.** There are multiple neuroendocrine cell types in the pancreas including alpha, beta, gamma, delta, and epsilon. We used gene expression data for these various pancreatic neuroendocrine and exocrine cell types from a single cell sequencing study[11] (Supplementary Table 2) to identify gene-set signatures representing highly expressed cell-type-specific genes (Supplementary data 4). The A-D-M mutant PanNETs uniformly exhibited a gene expression signature that was very similar to that of alpha cells (Fig. 4a). The A-D-M WT PanNETs were more heterogeneous in their expression of the genes among the gene set signatures for the different pancreatic neuroendocrine cell types. Greater heterogeneity of gene expression signature in A-D-M WT PanNETs was consistent with the greater heterogeneity found in global gene expression.

To further investigate the gene expression signature of A-D-M mutant PanNETs we performed gene set enrichment analysis[12] (GSEA) on the 13 manually curated gene sets for pancreatic endocrine and exocrine cells from a previous study. This study assessed gene expression of individual pancreatic cell

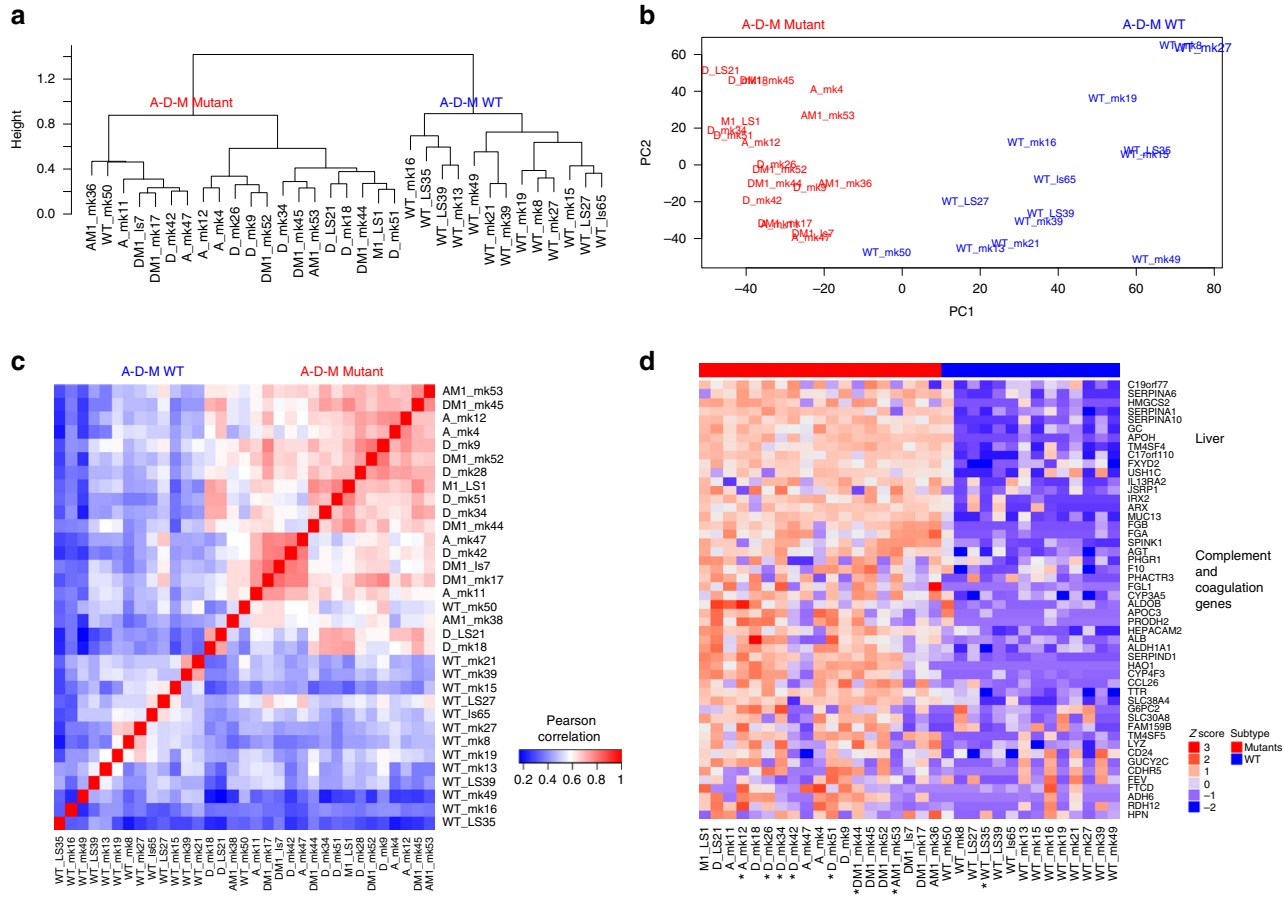

**Fig. 2** A-D-M mutant and WT PanNETs as two distinct gene expression groups. **a** Unsupervised clustering of PanNETs using top 3000 variant genes across all samples revealed two distinct robust clusters. The two clusters almost perfectly separate A-D-M WT panNETs from A-D-M mutant panNETs. **b** Principal component analysis using top 3000 variant genes separated the A-D-M mutant from A-D-M WT PanNETs along the first principal component (PC1). A-D-M mutant panNETs were more homogeneous in gene expression than A-D-M WT as shown by smaller variation along PC1. **c** Heatmap of pairwise Pearson correlation of panNETs using top 3000 variant genes across all samples revealed a higher correlation among A-D-M mutants as compared to A-D-M WT panNETs. Red color represents higher correlation and blue represents lower correlation. **d** Heatmap of top variants genes showing liver, complement, and coagulation genes highly expressed in A-D-M mutant panNETs. Star (*) below sample names represent liver metastatic samples (except for A_mk12 which is a lymph node)

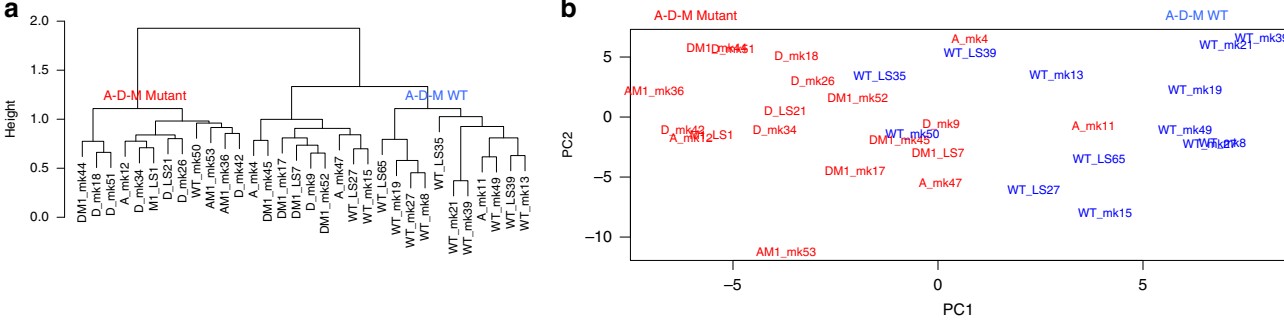

**Fig. 3** Distinct DNA methylation pattern between A-D-M mutant and A-D-M WT PanNETs. **a** Unsupervised clustering of PanNETs using top 2000 variant CpG sites across all samples revealed two clusters. The two clusters separate A-D-M mutant from A-D-M WT PanNETs. **b** Principal component analysis using top 2000 variant CpG sites separated A-D-M mutant from A-D-M WT PanNETs along PC1

types (alpha, beta, delta, PP, acinar, ductal, mesenchyme, and endothelial) enriched by flow cytometry and using single cell RNAseq (Supplementary Table 2). Our analysis indicates that only the alpha cell gene signature was significantly enriched in A-D-M mutant PanNETs (FDR $q$-value < 0.009) (Fig. 4b, c) (Supplementary Table 3).

Alpha and beta cell lineage specific genes were examined for the A-D-M mutant and WT PanNETs. *ARX*, *IRX2*, and *TM4SF4* were all highly expressed in A-D-M mutant PanNETs compared to A-D-M WT PanNETs (Supplementary Figure 7). Surprisingly, *GCG* (glucagon) expression was lower in A-D-M mutants as compared A-D-M WT PanNETs. For beta cell specific genes,

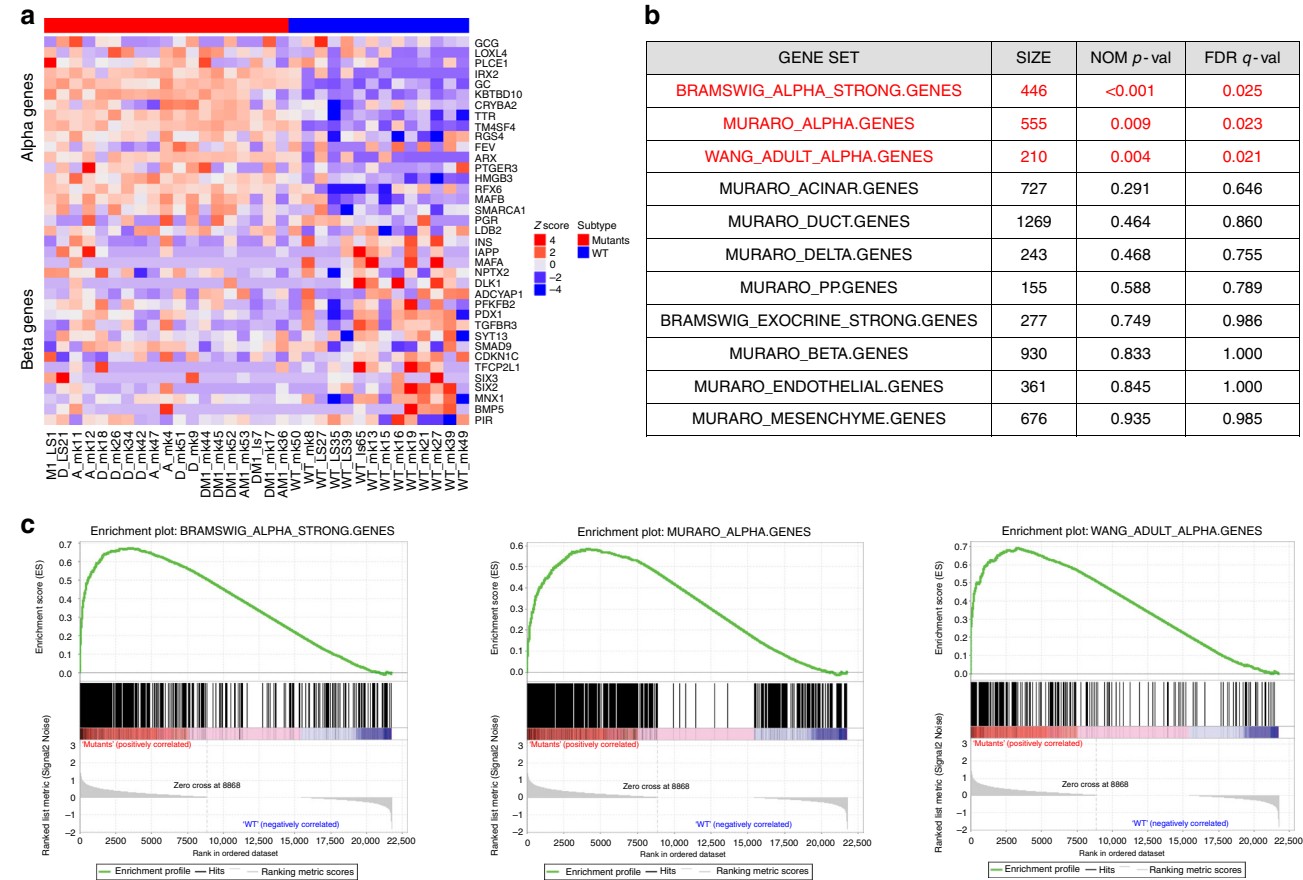

**Fig. 4** A-D-M mutant PanNETs with alpha-cell signature. **a** Heatmap of gene expression for top 20 alpha and beta cell-specific genes from Muraro et al.[11] revealed alpha cell specific genes are highly expressed in A-D-M mutant panNETs. A-D-M WT panNETs are more heterogeneous in gene expression but some show high beta cell specific gene expression. Red color represents higher correlation with alpha cell specific genes. **b** Gene set enrichment analysis show A-D-M mutant PanNETs to be enriched for expression of alpha cell specific genes. Pancreas cell type (alpha, beta, delta, PP, acinar, ductal) gene signatures were obtained from three different published dataset to access enrichment of cell type signatures in A-D-M mutant vs A-D-M WT PanNETs. Table represents GSEA results where size is the number of genes in gene set. All alpha cell gene sets (from three different sources) are significantly enriched in A-D-M mutant panNETs (highlighted in red). No other cell types were enriched in A-D-M mutant or A-D-M WT panNETs. **c** GSEA plots of significant alpha cell signatures from Bramswig et al.[15], Wang et al.[32], and Muraro et al.[11]

*PDX1*, *MAFA*, *INS*, and *DLK1*, all had lower expression in A-D-M mutant PanNETs than A-D-M WT PanNETs (Supplementary Figure 7). However, these genes had much greater expression heterogeneity in A-D-M WT PanNETs suggesting that some A-D-M WT PanNETs resemble beta cells and others did not (Supplementary Figure 7).

**Validation of subtype and alpha cell signature in A-D-M mutant PanNETs**. We derived an A-D-M mutant gene expression signature from significant differentially expressed genes between the A-D-M mutant and WT PanNETs from our data set ($n = 33$). We used two independent panNET[6,13] data sets to validate the gene expression signature of A-D-M mutant panNETs. We obtained A-D-M mutation status and gene expression dataset from International Cancer Genome Consortium Pancreatic Endocrine Neoplasm (ICGC PAEN) ($n = 29$)[6] and Sadanandam et al. ($n = 75$)[13]. The A-D-M mutant and WT PanNETs from both data sets have significant positive and negative correlations with our A-D-M mutant PanNET signature respectively (Fig. 5a) (Supplementary data 5). Additionally, we found alpha cell signatures to be significantly enriched (FDR q < 0.001) only in the A-D-M mutant PanNETs from the two validation data sets using GSEA (Fig. 5b, c).

**HNF1A pathway is up-regulated in A-D-M mutant PanNETs and alpha cells**. HNF1A is one of the most significantly differentially expressed genes between A-D-M mutant and WT PanNETs. HNF1A is a homeobox family transcription factor that is highly expressed in the liver and is involved in the regulation of several liver-specific genes. The expression of HNF1A was 2.93 fold higher in A-D-M mutant PanNETs than A-D-M WT PanNETs (corrected p-value < 0.004, Benjamini–Hochberg) (Fig. 6a). Differentially expressed genes (DEgenes) between the A-D-M mutant and A-D-M WT PanNETs were found in 1478 genes (with greater than 3 fold change and corrected *p*-value < 0.05 Benjamini–Hochberg, see Methods section) (Supplementary data 6). Functional pathway enrichment for DEgenes using pre-ranked GSEA revealed the complement and coagulation cascades, retinol metabolism, and drug metabolism to be up-regulated in A-D-M mutant PanNETs (see Methods section) (Fig. 6b; Supplementary data 6). The differentially expressed genes were also enriched for HNF1A transcription factor motifs in their promoters (FDR < 0.001, Fig. 6c). The complete list of significant TF motifs is presented in Supplementary data 6. Taken together, the A-D-M mutant PanNETs had higher expression of *HNF1A* along with many of its transcriptional target genes associated with liver function. In addition, the transcriptional regulator of *HNF1A*, *HNF4A*[14] was expressed 3.02 fold higher in A-D-M mutant

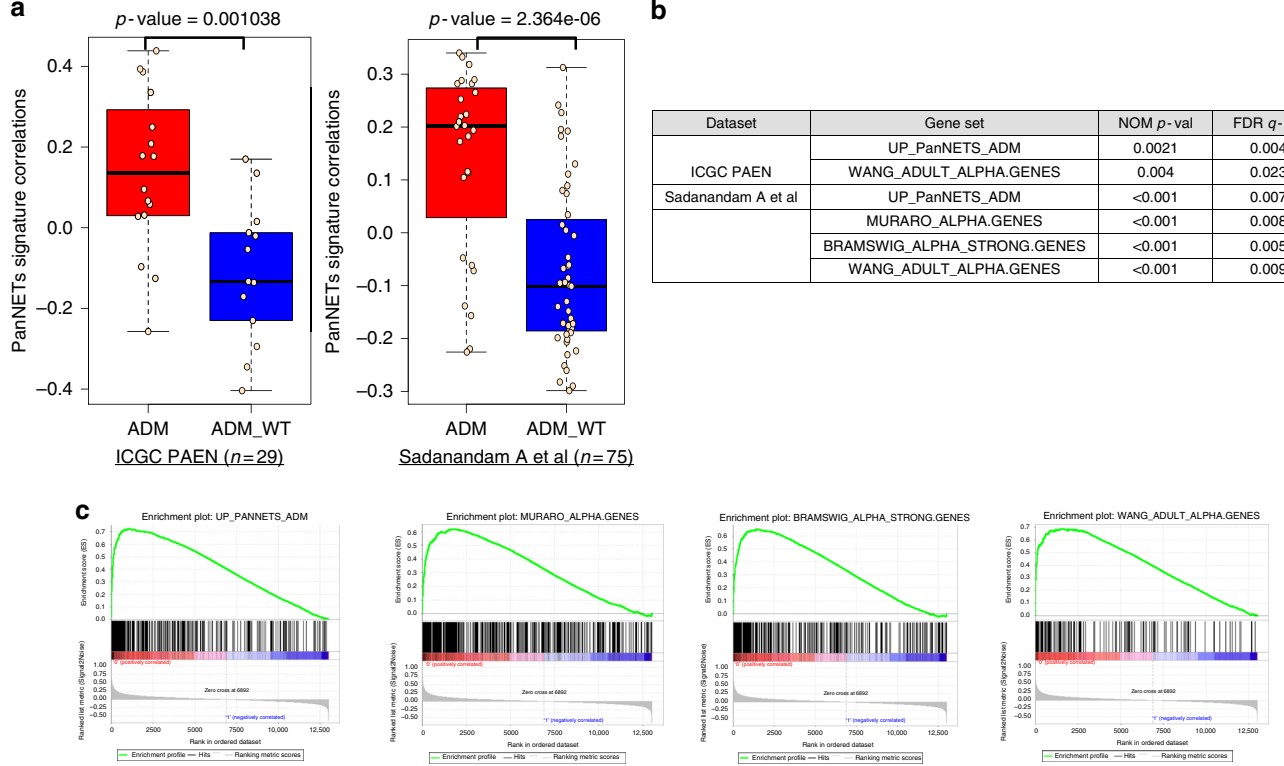

**Fig. 5** Validation of A-D-M mutant PanNET and alpha cell signatures. **a** Pearson correlation boxplot for two independent PanNET datasets show significant positive and negative correlations of A-D-M mutant and WT PanNETs with our A-D-M mutant PanNETs signature respectively (red represent A-D-M mutant and blue represent A-D-M WT with Wilcox p-value; center line is median, bounds of box are first and third quartile, and whiskers are min and max). **b** GSEA analysis shows A-D-M mutant PanNETs from ICGC PAEN[6] and Sadanandam et al.[13] are enriched for A-D-M mutant and alpha cell gene signatures. **c** GSEA enrichment plot for significant gene set for A-D-M mutant and alpha cell gene signatures from Sadanandam et al. dataset[13]

PanNETs ($p$-value < 0.009, DeSeq2). We used gene expression data from Bramswig et al.[15] to show *HNF1A* expression was increased in alpha cells compared to beta cells ($p$-value < 0.008), and the 465 alpha cell specific genes in the pancreas were enriched for transcriptional targets of *HNF1A* and for having *HNF1A* TF motif in their promoters (See Methods section; Supplementary Table 4).

Many of the most differentially expressed genes and highly expressed in A-D-M mutant PanNETs are targets of *HNF1A* and are involved in protein secretion, transport and metabolism (*APOH, ALB, AFM, HAO1, UGT1A3, UGT1A1, GC, G6PC, TM4SF4, PKLR* etc). *APOH* is expressed 8.46 fold higher in A-D-M mutant PanNETs ($p$-value < $10^{-5}$) and is potentially a good diagnostic biomarker for A-D-M mutant PanNETs. Moreover, *APOH* has been shown to have high expression in only the alpha cells of the pancreatic islet in a single cell RNA sequencing study[16]. We perform IHC staining for *APOH* and show positive staining in 70 ± 2.5% of A-D-M mutant and only 18 ± 2.0% of A-D-M WT PanNETs (Supplementary Figure 8).

***PDX1* gene is hypermethylated with low expression in A-D-M mutant PanNETs.** There is no genome wide hypo or hypermethylation of DNA in A-D-M mutant or WT panNETs. DNA methylation differences between the A-D-M mutant and A-D-M WT PanNETs were found at 378 CpG sites (corrected $p$-value < 0.05 and difference in beta value > 0.2, Benjamini–Hochberg), 287 of which were found in genes and 91 in intergenic regions (Supplementary data 7). Of the 287 differentially methylated genic CpG sites, 70 (associated with 59 genes) were found at promoter (transcriptional start site, TSS1500 and TSS200) or within first exon, a region where DNA methylation is associated

with transcriptional repression[17]. Thirteen of the 59 genes were also found to be differentially expressed (with fold change greater than 3 and corrected $p$-value < 0.05, Benjamini–Hochberg) and seven genes that were hypomethylated in A-D-M mutant and over-expressed are *APOH, CCL15, EMID2, PDZK1, HAO1, BAIAP2L2,* and *NPC1L1*. One gene, *TACR3*, was hypomethylated in A-D-M WT and over-expressed (Supplementary data 7). Four of the 70 CpG sites were found in the gene *PDX1* (pancreatic and duodenal homeobox 1), a transcription factor necessary for pancreatic development and beta cell maturation. *PDX1* functions in the cell fating of endocrine cells, favoring the production of insulin positive beta cells and somatostatin positive delta cells while repressing glucagon positive alpha cells[18]. These four CpG sites were all hypermethylated in A-D-M mutant PanNETs (Fig. 7a) and the expression of *PDX1* was 2.92 fold higher in A-D-M WT PanNETs ($p$-value < 0.005, DeSeq2) (Fig. 7a, b). In contrast, while *ARX* was highly expressed in A-D-M mutant PanNETs compared to A-D-M WT PanNETs, the promoter and first exon of ARX are not differentially methylated.

## Discussion

Similar to a number of recent studies[19,20], we have demonstrated in this cohort of PanNETs that, in additional to pathologic stage and grade of the tumor, mutations in *DAXX, ATRX,* and *MEN1* are associated with adverse clinical outcome in comparison to those without these mutations. Our results seem to be in contradiction to the findings initially reported by Jiao et al.[5], in that 15 patients with PanNETs carrying mutations in *DAXX* or *ATRX* genes had better survival than did 12 patients with wild-type PanNET. This discrepancy between our data and their data could be attributed to a different composition of the tumors. Indeed, all

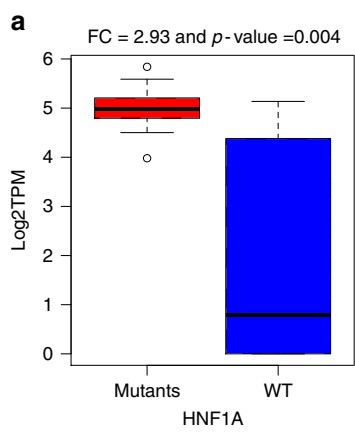

**a** FC = 2.93 and *p*-value =0.004

**b**

| Subtype | NAME | SIZE | FDR *q*-val |
|---------|------|------|-------------|
| Mutant | COMPLEMENT_AND_COAGULATION_CASCADES | 24 | <0.001 |
| Mutant | RETINOL_METABOLISM | 25 | <0.001 |
| Mutant | DRUG_METABOLISM_CYTOCHROME_P450 | 21 | <0.001 |
| Mutant | METABOLISM_OF_XENOBIOTICS_BY_CYTOCHROME_P450 | 21 | <0.001 |
| Mutant | STEROID_HORMONE_BIOSYNTHESIS | 15 | <0.001 |

**c**

| Subtype | Motif NAME | SIZE | FDR *q*-val |
|---------|-----------|------|-------------|
| Mutant | HNF1_Q6 | 44 | <0.001 |
| Mutant | HNF1_01 | 47 | <0.001 |
| Mutant | HNF1_C | 40 | <0.001 |
| Mutant | TTANTCA_UNKNOWN | 20 | <0.001 |

**Fig. 6** *HNF1A* pathway with transcriptionally up-regulation in A-D-M mutant panNETs and alpha cells. **a** Boxplot of *HNF1A* gene expression for A-D-M mutant and A-D-M WT PanNETs. *HNF1A* was homogeneously expressed 2.93 fold higher in A-D-M mutants PanNETs (corrected *p*-val < 0.004, Benjamini–Hochberg). Center line, median; bounds of box, the 1st and 3rd quartiles; and upper and lower whisker is defined to be 1.5 × IQR more than the third and first quartile. **b** Table represents significant KEGG pathways where genes were differentially expressed between A-D-M mutants and A-D-M WT panNETs. **c** Table represents transcription factor motifs significantly enriched in promoters of genes differentially expressed in A-D-M mutants and A-D-M WT panNETs. Three HNF1 related motif gene sets from GSEA showed significant enrichment in genes over-expressed in A-D-M mutant panNETs. GSEA was used to find pathway enrichment from genes differentially expressed between A-D-M mutant and A-D-M WT PanNETs

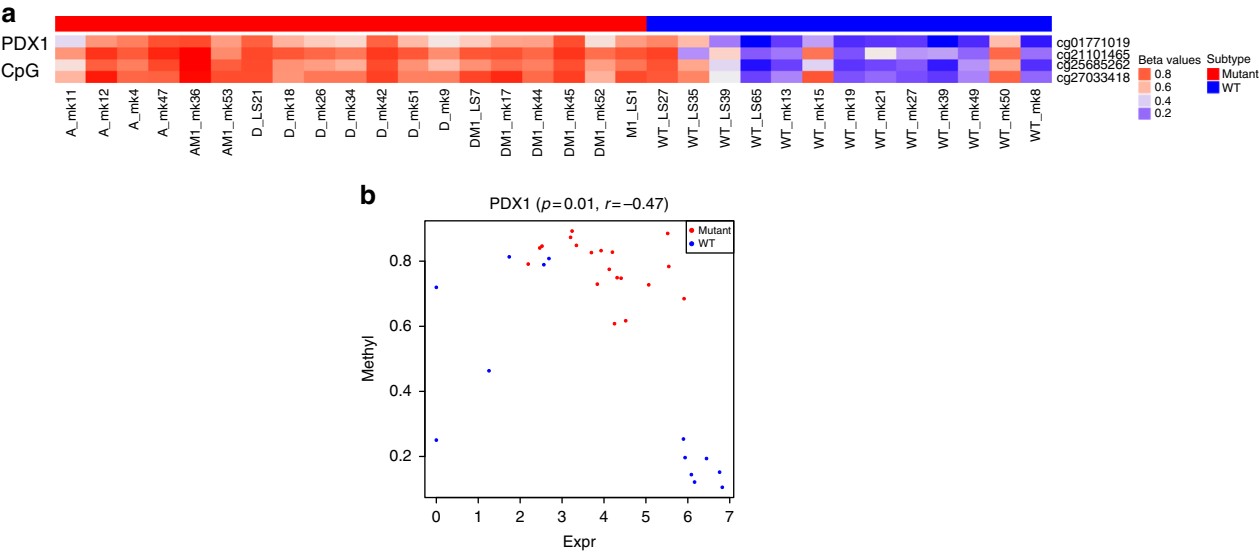

**Fig. 7** PDX1 has promoter hypermethylated and lower gene expression in A-D-M mutant panNETs. **a** Four PDX1 promoter CpG sites show strong hypermethylation in A-D-M mutant PanNETs (corrected *p*-val < 0.05, Benjamini–Hochberg). The range of beta values is from 0 to 1 and represented as blue (hypo-methylation) to red (hyper-methylation). **b** PDX1 expression and promoter methylation (TSS1500 cg27033418 CpG site) across all samples showing separation of A-D-M mutant and A-D-M WT PanNETs

the tumors analyzed in Jiao's study[5] were liver metastases from PanNETs as opposed to only 19 % (12 out of 64) in our study. Other factors including sample size and length of follow up time may also contribute to discrepancies between different studies.

Here, we found that A-D-M mutant PanNETs form a distinct subgroup on the basis of their gene expression profile and DNA methylation pattern. Moreover, this subgroup is more homogeneous based on gene expression profile than the A-D-M WT PanNETs. The gene signature of the A-D-M mutant PanNETs strongly corresponds to the genes that are specifically expressed in alpha cells including genes known to define alpha cells such as *ARX* and "liver-specific" genes such as *HNF1A* and its transcriptional targets. Conversely, *PDX1*, a gene critical to the beta

cell lineage is transcriptionally repressed in A-D-M mutant PanNETs and the *PDX1* promoter is hypermethylated. On the other hand, WT PanNETs have heterogeneous gene expression profiles and their gene mutational landscape is less understood.

The pancreas is comprised of many different cell types including acinar, ductal, and at least five neuroendocrine cell types including alpha, beta, gamma, delta, and epsilon cells. There are two plausible explanations for the "alpha cell-like" expression pattern of A-D-M mutant PanNETs. Either an alpha cell or an uncharacterized cell type with an alpha-cell like gene expression profile is the cell-of-origin for PanNETs with mutations in *ATRX*, *DAXX*, and *MEN*, or loss of *ATRX*, *DAXX*, or *MEN1* genes may promote pancreatic neuroendocrine (or progenitor) cell types to

reprogram their gene expression profiles to resemble alpha cells. It remains unclear whether there are pancreatic stem or progenitor cells in adult pancreas.

*ATRX-DAXX* and *MEN1* are involved in distinct biochemical pathways to regulate gene expression. Therefore, we would expect that loss of these proteins during transformation of A-D-M mutant PanNETs would result in a more heterogeneous gene expression profile. Due to the high degree of homogeneity of the A-D-M mutant PanNETs at the level of gene expression and the strong expression of genes that are known to be alpha cell specific, we hypothesize that alpha cells are the cell-of-origin for this group of tumors. In addition, *MEN1* and *ATRX/DAXX* mutations occur alone or in a combined pattern suggest that they have independent oncogenic activities in A-D-M mutant PanNETs, making the idea of reprogramming to a homogeneous alpha-like cell state less probable. Some of the A-D-M WT PanNETs have a strong beta cell signature and these may have arisen from beta cells (Fig. 4a). However, other A-D-M WT PanNETs have neither alpha nor beta cell signatures, which may arise from other cell types in the pancreas.

Conditional knockouts of *MEN1* in mice support the model of an alpha cell origin for A-D-M PanNETs[21]. The restricted deletion of *MEN1* to alpha cells surprisingly led to the development of insulinomas[22,23]. Most of our PanNETs were nonfunctional (26 of 33 PanNETs) but the functional tumors were insulinoma and VIPoma, even though their gene expressions have alpha cell signature. Moreover, some PanNETs express combinations of neuroendocrine hormones (*GCG, INS, SST, PPY, GHRL, VIP,* and *GAST*), suggesting that regulation of cell type specific hormone may be disrupted. To create robust gene signatures that are not sensitive to changes in expression of a few genes, we use a large number of genes to create the A-D-M mutant and alpha cell signatures. In other mouse models of PanNETs[21,24,25], *MEN1* deletion using the insulin or *PDX1* promoter driven Cre construct, insulinomas, glucagon-expressing tumors and well differentiated PanNETs were also observed. However, Cre expression may be leaky in these models and further study is needed to understand the heterogeneity of the cells in the tumors that develop and trace the cell of origin or transdifferentiated state of the cancer cells.

In our gene expression analysis, we have not identified the oncogenic pathways activated in A-D-M mutant PanNETs. *MEN1* has been shown to upregulate expression of long noncoding RNA *MEG3* in MIN6 mouse insulinoma cell line[26]. In the same study, they show *MEG3* represses expression of the oncogene MET leading to delayed cell cycle progression and reduced cell proliferation. In a different study, *MEN1* and *DAXX* were shown to repress the expression of the membrane metalloendopeptidase (*MME*) and mutations in *MEN1* or *DAXX* result in loss of this repression leading to neuroendocrine tumor proliferation[27]. Our data is consistent with these studies when comparing A-D-M mutant to WT PanNETs, showing that A-D-M mutant PanNETs have lower expression of *MEG3* (7.3 fold lower, *p*-value < 4.3E-07, DeSeq2), higher expression of *MET* (3 fold higher, *p*-value < 0.003, DeSeq2), and higher expression of MME (4 fold higher, *p*-value < 0.001, DeSeq2). Among A-D-M mutant PanNETs, we do not see expression differences of *MEG3, MET,* and *MME* depending on mutation status of *ATRX, DAXX,* and *MEN1*.

While PanNETs may seemingly represent as a single clinical disease, they can be further characterized into different subtypes based upon their cell lineage and the associated molecular genotype. Understanding the epigenetic and transcriptional dysregulation of PanNETs will require comparison to their proper cells of origin which may explain the unpredictable outcome of the disease and facilitate the development of unique and targeted therapeutic strategies.

## Methods

**Patient's information**. Retrospective and prospective reviews of well-differentiated, pancreatic neuroendocrine neoplasms were performed using the pathology files and pancreatic database at MSKCC with IRB approval. All patients were evaluated clinically at our institution with confirmed pathologic diagnoses, appropriate radiological and laboratory studies, and surgical or oncological management. Follow-up information was obtained for all cases.

**Tissue acquisition and nucleotide extraction**. Briefly, cases of pancreatic neuroendocrine tumors were identified. Fresh-frozen tumor and paired normal tissues were obtained from MSKCC's tissue bank under an Institutional Review Board protocol. Histopathology of all tissues was evaluated on hematoxylin and eosin stained sections by an experienced gastrointestinal-hepato-pancreatobiliary pathologist to insure the nature of the tissue, greater than 80% tumor cellularity and absence of necrosis. The relevant tissues were then macro-dissected (20–25 mg) and DNA/RNA extraction using Qiagen's DNeasy Blood & Tissue Kit and RNeasy Mini Kit, respectively was carried out according to the manufacturer's protocols (Qiagen, Valencia, CA).

**Sanger sequencing for gene mutation**. All exons of the *DAXX, ATRX,* and *MEN1* genes were amplified by PCR and then sequenced using Sanger sequencing. Every mutation detected was validated by bidirectional Sanger sequencing on the tumor-normal pairs. To maintain the correct sample annotation, we used mutation status as sample name with sample ID (for example, A_mk11 sample is *ATRX* mutant and mk11 is sample ID). Supplementary data 1–3 contains all the clinical information, mutational profile, sample annotation and ESTIMATE tumor purity. Online Oncoprint was used to plot create Fig. 1a.

**PanNETs transcriptome sequencing and data analysis**. RNA Library preparation and RNA sequencing was done by MSKCC Genomics Core Laboratory using Illumina HiSeq with (2 × 75 bp paired end reads) to a minimum depth of ~50 million reads were generated for each sample. Raw fastq files were probed for sequencing quality control using FastQC [http://www.bioinformatics.babraham.ac.uk/projects/fastqc]. Sequencing reads were mapped to human transcripts corresponding to Genepattern[28] genome (hg19 version) GTF annotations using RSEM with default parameters. RSEM package[29] was used to prepare the reference genome with given GTF and calculated expression from mapped BAM files. STAR[30] aligner was used to map reads in RSEM algorithm. Transcripts mapped data were normalized to TPM (Transcript Per Million) from RSEM and log2 transformed (Supplementary data 8). This log2TPM values were used for all downstream analysis. Unsupervised clustering and Principal Component analysis was conducted to elucidate subtypes structure using top 3000 variant genes as input. To query robustness of this subtyping, multiple variant gene sets were used and repeated the same process of unsupervised clustering. Top 100 variable genes were used to find genes, which were highly expressed in each subtype. Subset of these genes is selected to show in Fig. 2d for liver and complement system genes. To find differentially expressed genes (DEgenes) between A-D-M mutant PanNETs and A-D-M WT panNETs, we used DeSeq2 R package[31] on raw count (values from RSEM). We used significance cutoff with greater than 3 fold change and corrected *p*-value < 0.05 (Benjamini–Hochberg) to call a gene as DEgenes. GSEA Preranked[12] method was used on DEgenes to find significant KEGG pathways, motif and biological process.

**Clustering and principal component analysis**. For unsupervised clustering on log2TPM, we used Pearson distance metric and ward.D2 hclust method (unless stated otherwise). PCA analysis was done using *prcomp* in R. R (http://www.r-project.org/) was used for all the analysis and visualization of data.

**PEEGset from published dataset**. The neuroendocrine cells in the pancreas include alpha, beta, delta, pancreatic polypeptide (pp)-producing and vasoactive intestinal peptide (VIP)-producing cells. Gene sets representing different endocrine islet and exocrine pancreatic cells (PEEGset) were obtained from three metadata[11,15,32] (Supplementary Table 2). We created 13 PEEGset representing all major cells from endocrine and exocrine pancreases. Supplementary Table 2 shows these gene sets with major cell types and number of genes in each set. These gene set were used as prior defined gene set for GSEA analysis.

**Gene set enrichment analysis on major islets cell types**. Gene Set Enrichment Analysis[12] (GSEA) was performed on the log2TPM expression values of all samples using downloaded version of GSEA software (Broad Institute, Cambridge, MA, USA) to identify the statistically enriched gene sets between A-D-M mutant and A-D-M WT PanNETs. Published pancreatic islet endocrine and exocrine cells signatures were used as prior defined sets as an input. We used all default parameters to perform GSEA on this gene sets to determine the enrichment of specific cell signature enrichment in the PanNET subtypes. We ran GSEA on 1000 permutation mode on phenotypic label to generate FDR and enrichment score (ES) for each gene set. Significant gene set was filtered based on FDR q-values (cutoff of 0.05).

**Bramswig et al. FACs sorted alpha and beta cells gene expression.** We extensively used Bramswig et al.[15] FACs sorted RNAseq data to understand normal alpha and beta cells and correlated their gene signature sets with our A-D-M mutant and A-D-M WT panNETs. We downloaded supplement file for total RNA seq normalized expression data for alpha (3 replicate) and beta (3 replicate) and exocrine cells (2 replicate). Bramswig et al.[15] provide strong genes associated with alpha, beta and exocrine cells as supplement file. We used this strong cell specific genes and created gene set for alpha, beta and exocrine and named as Bramswig et al. gene set. *HNF1A* gene expression values were fetched to check whether HNF1A is over expression in normal alpha as compared to beta and exocrine. We applied Student ttest's between three alpha and three beta samples to calculate *p*-value for *HNF1A* gene expression. Bramswig et al. strong alpha cell genes ($n = 465$) were queried to check for *HNF1A* transcription factor motif enriched using online GSEA version (C3 TFs motif database).

**Microarray gene expression analysis.** RNA extracted from PanNETs samples were submitted for Affymetrix microarray profiling using chip hgu133a2 array. All following analysis was done in R. Briefly, the raw Affymetrix CEL files were loaded in R (simpleaffy[33]). Normalized expression values were calculated by the GC Robust Multi-array Average (GCRMA[34]) algorithm and subjected to mean transformation in order to collapse all probes to respective genes in R using collapseRow[35]. We then followed the same clustering and PCA procedure that were done on RNAseq expression data. Collapsed average gene expression values were imported in GSEA to run against 13 PEEGset cell types to find enrichment.

**450 K DNA methylation array analysis.** DNA extracted from PanNETs samples and interrogated using the Illumina 450 K platform (Illumina Inc. San Diego, CA) to access the DNA methylation profiles. All the analysis was performed using ChAMP[36] version 2.6.0 open source software implemented in R. Briefly, IDAT file raw data were imported in R and filtered to exclude samples with detection *p*-value < 0.01 (DeSeq2) and beadcount <3 in at least 5% of samples and normalized using FunctionNormalization[37]. This normalization method correct for background; remove dye bias followed by Quantile normalization. Unsupervised clustering and PCA were done on top variants 2000 probes (Var2000) across all samples to find classes of PanNETs. We repeated this clustering using different number (Var10000, Var5000, Var3000, Var1000, and Var500) of probes to check robustness of this subtyping. Differentially methylated CpG sites (DMP) between the A-D-M mutant and A-D-M WT PanNETs were identified using champ.MVP using the all default parameter method (Bonferroni-Hochberg) to adjust the *p*-value(<0.01, DeSeq2). Significant DMP sites from respective genes were compared to DEgenes to find overlapping dysregulated genes in each subtype.

**A-D-M mutant PanNET Signature and validation.** Significant differentially expressed genes (fold change ≥3 and Corrected Pval <0.05, Benjamini–Hochberg) between A-D-M mutant and WT panNETs were used with log2 transformation of fold changes to create an A-D-M mutant PanNETs signature for validation. We downloaded gene expression and genotype data from two independent PanNETs cohort (a) ICGC Pancreatic Cancer Endocrine Neoplasms[6] (PAEN)(ref nature and ICGC site) and (b) Sadanandam et al.[13]. ICGC PAEN processed A-D-M mutation status and RNAseq gene expression dataset (FPKM normalized expression) were downloaded from ICGC website (http://icgc.org/icgc/cgp/68/304/1003406) for 29 samples (16 A-D-M Mutant and 13 A-D-M WT). Sadanandam et al.[13] performed targeted sequencing of *ATRX, DAXX, MEN1, PTEN, TSC2* and *ATM* on 75 PanNETs. Genotype information was downloaded from supplementary data of [13]PMID: 26446169 and matched microarray gene expression data for 75 PanNETs (28 A-D-M and 47 A-D-M WT) were downloaded from NCBI GEO (Accession Number GSE73338). Normalized gene expression values obtained from GSE73338 were used for gene signature analysis. Pearson correlation was calculated between our A-D-M mutant gene signature and the mean-variance normalized gene expressions from ICGC PAEN and Sadanandam et al.[13] and pval was calculated using Wilcox test. We performed GSEA analysis on A-D-M mutant and WT panNETs from Sadanandam et al.[13].

**Immunohistochemistry (IHC).** A representative, formalin-fixed, paraffin-embedded tissue section (4 μm thick) of each case was submitted to our institution's core facility to perform immunohistochemistry-using antibodies recognizing the APOH proteins. Briefly, sections were de-paraffinized and pre-treated in Cell Conditioning 1 (CC1 mild; Ventana Medical Systems, AZ, USA) using an automated staining system (Ventana Discovery XT Autostainer; Ventana Medical Systems Inc, Tucson, AZ). Primary antibodies were applied for 60 min at a dilution of 1:100 for *APOH* (anti-APOH, polyclonal antibody; Proteintech). The sections were then incubated for 60 min with secondary antibody (1:200) followed by DAB Map detection (DAB visualization; Ventana Medical Systems). Cytoplasmic (*APOH*) labeling in at least 50% of the tumor cells was considered positive. In the case of *APOH*, normal liver tissue was used as a positive control in each experiment.

**Histone marks IHC.** Serial unstained slides (4 μm) were prepared from each block for subsequent immunohistochemistry with the following Histone 3 lysine antibodies cones: H3K4me3 clone C42D8 (Cell Signaling Technologies Catalog number 9751 1:1000 dilution), H3K9me3 clone EPR16601 (abCam, Catalog number Ab8898 1:1000 dilution), H3K27me3 clone C36B11 (Cell Signaling, Catalog number 9733 1:100 dilution) and H3K36me3, clone 333 (Active motif, Catalog number 61021, 1:500 dilution). Staining for all clones was performed on the Leica Bond immunohistochemistry platform according to the manufacturer's protocol with the Lieca DAB IHC detection kit. All slides were pretreated with epitope retrieval 2 solution (Lieca Biostems) for 30 min. Primary antibodies were incubated for 30 min. Multi-tissue normal positive control was used. The PanNET cases ($n = 36$, 14 A-D-M mutant and 22 A-D-M WT) were read and interpreted by an independent observer blinded to the clinicopathologic information. Scoring of all histone marks was performed using previously validated scoring systems H3K4me3[38], H3K9me3[39], H3K27me3[40], H3K36me3[41]. The tumor was considered positive for the histone mark if there was histological evidence of nuclear staining. Every tumor was scored on a scale of 0–3 according to the percentage of cells with nuclear staining: (0, 0–5% positive cells; 1, 6–50% positive cells; 2, 51–75% positive cells; 3, 76–100% positive cells). Scores of 0–1 were estimated as low expression and scores of 2–3 indicated high expression. Student t-test was used to test significance for histone methylations across *MEN1* panNETs as compared to *MEN1* WT PanNETs.

**Statistical analysis.** Data are represented as mean ± standard deviation. GraphPad Prism 6 (GraphPad Software Inc, La Jolla, Ca) was used for statistical and survival analyses. Survival analysis *p*-values (2-sided) were based on log-rank tests. Significance was defined as $P < 0.05$.

## Data availability

The authors declare that all data supporting the findings of this study are available within the article and its Supplementary data and figures. Data generated in this study were deposited to NCBI under GEO SuperSeries GSE117853 (GSE118014 for RNAseq, GSE117851 for Microarray and GSE117852 for 450 K methylation). We do not impose any restrictions on data availability.

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

## Acknowledgements

We would like to thank Starr Cancer Consortium Grant to (CDA and LHT), Raymond and Beverly Sackler Foundation (LHT), Caring for Carcinoid Foundation (CDA and LHT), Mushett Family Foundation (LHT), MSKCC Support Grant/Core Grant (P30 CA008748). This research was also supported by the Biomedical Informatics shared resource of Rutgers Cancer Institute of New Jersey (P30CA072720). Computational resources were provided by the Office of Advanced Research Computing (OARC) at Rutgers, The State University of New Jersey, under the National Institutes of Health Grant No. S10OD012346.

## Author contributions

Chang S Chan: computational design, computational data analysis, manuscript preparation. Saurabh V. Laddha: computational data analysis, manuscript preparation. Peter Lewis: experimental design, implementation, and manuscript review. Matthew Koletsky: experiment implementation. Kenneth Robzyk: data discussion and manuscript review. Edaise Da Silva: specimen preparation and clinical data collection. Paula J. Torres: specimen preparation. Brian R. Untch: data discussion and manuscript review. Janet Li: experiment implementation. Promita Bose: data analysis. Timothy A. Chan: data discussion and analysis. David S. Klimstra: pathology advisor. C. David Allis: study design and scientific advisor. Laura H. Tang: study design, data analysis, manuscript preparation.

## Additional information

**Competing interests:** The authors declare no competing interests.

