## [Peer Review File · Nature Communications]

Reviewers' comments:

Reviewer #1 (Remarks to the Author):

Several studies have demonstrated that a large portion of PanNETs have mutation in MEN1 and/or ATRX/DAXX, but a large proportion does not have mutations in these genes. The primary PanNETs without mutations in these genes still metastasize but overall they have better prognosis. It has been hypothesized that the cell of origin is different between the PanNETs with mutation in these genes, called here A-D-M and WT. This study suggests that the cell of origin for A-D-Ms is the alpha pancreatic cell based on expression profiles.

Comments:

- This study provides preliminary data that the alpha type is the cell of origin for PanNETs with mutations in MEN1 and/or ATRX/DAXX. However, there is not any classifier presented here that can be used for this purpose and the expression profile is not validated well in an independent set of samples.
- The text indicates that tumor purity is 80%, but this is a pathological estimate. It will be good to see percent mutant in the samples that have mutations. It will be a more accurate way to determine purity. The upregulation of liver specific genes is curious and contamination needs to be excluded the best way possible.
- The fraction of the samples with DAXX or ATRX mutations appear lower than what has been previously published. Depending on the method for mutation detection, some mutations will be missed. Have the samples negative for mutations been checked for ALT, as a surrogate for ATRX/DAXX mutations, or for the absence/presence of ATRX or DAXX with an antibody? This will ensure that the WT group is indeed WT.
- It is not clear why 33 samples used for the first round of experiments and only additional 14, instead for the rest of them, for validation.
- It is concerning that cell type specific markers do not show a pattern consistent with the hypothesis that A-D-M come from alpha cells. For example GCG expression, a marker for alpha cells (Murano et al; Lawlor et al), is not specific for this group. Similar situation for INS, a beta cell marker, should not be expressed in this group.
- It will be interesting to see the classification using supervised analysis on mutations.

Reviewer #2 (Remarks to the Author):

The most interesting part of the study is that the data of RNA sequencing and DNA methylation assay showed that A-D-M mutant PanNETs have a distinct gene expression and DNA methylation patterns from that of A-D-M WT PanNETs, suggesting the importance of ATRX, DAXX and MEN1 in regulating PanNETs. The authors, using the single cell sequencing study, also found that the gene expression profile of the A-D-M mutant PanNETs strongly correlates to the profile of the genes that are specifically expressed in alpha cells, including genes known to define alpha cells such as ARX. In general, understanding of the origin PanNETs, tumor progression and pathways is important as it may help to detect novel targets for PanNET treatment. This study has the potential to explain the unpredictable outcome of the PanNETs and facilitate the development of unique and targeted therapeutic strategies. However, some concerns are included as follows and must be addressed:

1. Figure 1b clearly showed that among 44 patients who initially presented with localized PanNETs (without distant metastasis), those with A-D-M mutant genotype had a worse recurrence free survival outcome than those with A-D-M WT genotype in their primary tumors. How about the survival outcome of the PanNETs with A-D-M mutant and metastasis?
2. HNF1A, ARX, IRX2, and TM4SF4 were highly expressed in A-D-M mutant PanNETs compared to A-D-M WT PanNETs. However, did the authors perform the qRT-PCR or Western Blot to confirm that ATRX, DAXX and MEN1 indeed regulate these genes using the PanNET cell lines? This would

provide significant information regarding the functional importance of this correlation.

3. The authors did DNA methylation analysis and found that DNA methylation is distinct between the A-D-M mutant and A-D-M WT PanNETs. It is well documented that menin is very crucial for epigenetic regulation by histone modifications, such as H3K4me3 and H3K9me3. Therefore, whether the histone modifications are different between the A-D-M mutant and A-D-M WT PanNETs and whether the histone modifications contribute to the change of gene expression and DNA methylation? A functional and mechanistic study will surely strengthen the manuscript.

4. High ARX gene expression was observed with the promoter and first exon of ARX not differentially methylated, while low PDX1 gene expression was observed with PDX1 promoter hyper-methylation in the A-D-M mutant PanNETs as compared to A-D-M WT PanNETs. However, do ATRX, DAXX and MEN1 promote or repress the DNA methylation? Which is caused by ATRX, DAXX and MEN1 mutation? Certain cellular and functional studies would much improve the manuscript.

5. The frequencies of ATRX, DAXX and MEN1 mutations are 8%, 16% and 20%, respectively. Are these mutations found previously? Are these mutations reported to correlate to alternation of DNA methylation? A discussion about this would be of help.

Minor:

1. Lane 47, 222, 223...: MEN1 is the name of multiple endocrine neoplasia type1 and should be replaced by italic MEN1 or menin, the protein name of MEN1.

2. Lane 145, two "that".

3. Lane 240-241: "Alpha cell-specific Men1 ablation triggers the transdifferentiation of glucagon-expressing cells and insulinoma development" from Zhang CX group also found that deletion of MEN1 in alpha cells developed insulinomas.

4. "Conditional deletion of Men1 in the pancreatic β -cell leads to glucagon-expressing tumor development" from Guang Ning group found that specific Men1 knockout in β -cell developed both glucagon-expressing tumors and insulinomas. These glucagon-expressing tumor cells were menin deficient and expressed the mature α -cell-specific transcription factors Brain-specific homeobox POU domain protein 4 (Brn4) and v-maf musculoaponeurotic fibrosarcoma oncogene family, protein B (MafB). This reference should be included.

Reviewer #3 (Remarks to the Author):

This manuscript described a subset of pancreatic neuroendocrine tumors by three gene mutations, supported by gene expression and DNA methylation profiles, as a distinct alpha-cell subgroup, which appears having a clinical implication (i.e., patient survival). Although valuable to the medical/research community, the writing and analyses are a bit shallow. Lack of an independent validation is another concern for its certainty. Below are some specific points:

1) The background did not provide sufficient information in the topic. What other mutations are commonly found in this tumor? What are known predictors for clinical outcomes or classification methods for patient management (the authors mentioned grade and stage but did not say their clinical relevance)?

2) The method or result did not provide sufficient details about the mutations in these three genes. Do patients have the exact same mutation across individuals for the same category of mutation (missense, in-frame or truncating...)? For example, in-frame mutations are most common in this set of patients but are they exactly the same (at the same location, same length, insertion or deletion)? The supplementary data seems not including the detailed mutation information (or not uploaded).

3) In addition to the survival, authors should also correlate to other clinical variables such as grade and stage. Multivariable survival analysis is also needed to see if the classification by the genotype makes difference.

4) The authors should conduct comparisons with public data such as the two cited (ref 2,3). In Jiao's paper, they found the tumors with mutations in these three genes were associated with better survival but here it is reverse so further exploration and explanation are needed.

5) The integrative analysis between methylation and gene expression is weak. What is the set of genes that are likely impacted by DNA methylation and what is the set of genes that are not affected (in the mutation context)? Is the mutant group more hyper or hypo-methylated? Where are the changes mostly located in the genome (CpG Island, shore, shelf, intergenic region...)? Do the mutations affect DNA methylation? I guess the main question is whether the association is causal or coincidence.

Reviewers' comments:

We thank the reviewers for their valuable and critical comments and have made substantial changes to our manuscript. We will address their remarks point by point below.

Reviewer #1 (Remarks to the Author):

Several studies have demonstrated that a large portion of PanNETs have mutations in MEN1 and/or ATRX/DAXX, but a large proportion does not have mutations in these genes. The primary PanNETs without mutations in these genes still metastasize but over all they have better prognosis. It has been hypothesized that the cell of origin is different between the PanNETs with mutation in these genes, called here A-D-M and WT. This study suggests that the cell of origin for A-D-Ms is the alpha pancreatic cell based on expression profiles.

Comments:

- This study provides preliminary data that the alpha type is the cell of origin for PanNETs with mutations in MEN1 and/or ATRX/DAXX. However, there is not any classifier presented here that can be used for this purpose and the expression profile is not validated well in an independent set of samples.*

Response:

We thank the reviewer for this important comment and suggestion and have now added a new **Figure 5 and a result section**, titled “**Validation of distinct subtype and alpha cell signature in A-D-M mutant PanNETs**” (Page 6), and presented validation of our gene expression signature for A-D-M and WT PanNETs in two independent data sets^{1,2}. Specifically, we have demonstrated that the gene expression signature can separate the A-D-M mutant from WT PanNETs, and the A-D-M PanNETs are enriched for alpha cell signature.

- The text indicates that tumor purity is 80%, but this is a pathological estimate. It will be good to see percent mutant in the samples that have mutations. It will be a more accurate way to determine purity. The upregulation of liver specific genes is curious and contamination needs to be excluded the best way possible.*

Response:

We agree with the reviewer that assessing the tumor purity from allele frequency of mutations can often be a more accurate estimation. However, in our study, we only performed Sanger Sequencing for *MEN1*, *ATRX*, and *DAXX* so we are not able to impute accurate tumor purity from mutation allele frequency. Nevertheless, we used gene expression from RNA sequencing to estimate tumor purity with a widely used algorithm called ESTIMATE³. ESTIMATE and showed tumor purity to be greater than 80% for all samples except for two (DM1_mk44 has 73% and WT_mk13 has 43% tumor purity). These estimates of

tumor purity from ESTIMATE are consistent with the pathology estimates. High tumor purity can often be obtained from well differentiated PanNETs because of their unique histopathologic characteristics of hypercellularity, stromal-poor, and lack of tumor necrosis. In the study by Scarpa et al¹, they also estimated the median tumor purity of their PanNETs (103 samples) from mutation allele frequency to be 87%. We are including the ESTIMATE score for tumor purity in supplementary file 1c and refer to its use in Results section part 2 in the revised manuscript (Page 5).

The upregulation of “liver specific” genes are seen in both primary and unpaired liver metastasis of A-D-M PanNETs. Figure 2d shows heatmap of the top variants genes (including “liver specific” genes) for all samples with star(*) annotating the liver metastases. We believe liver tissue contamination in our primary PanNETs is nil since these specimens were collected in the absence of any liver tissue. We believe that those “liver specific” genes may represent dysregulated genes in A-D-M mutant PanNETs rather than genes of mature hepatic differentiation.

• The fraction of the samples with DAXX or ATRX mutations appear lower than what has been previously published. Depending on the method for mutation detection, some mutations will be missed. Have the samples negative for mutations been checked for ALT, as a surrogate for ATRX/DAXX mutations, or for the absence/presence of ATRX or DAXX with an antibody? This will ensure that the WT group is indeed WT.

Response:

We agree with the reviewer that the reported mutation rates of *DAXX* and *ATRX* in pancreatic NETs are not entirely consistent in the literature. The rate of *DAXX* mutation (including *MEN1/DAXX* double mutations) in the current study of 25% is identical to that in the initial report⁴; and the rate of *ATRX* mutation of 11% (including *MEN1/ATRX* double mutations) is slightly lower than previously reported 13%⁴.

In addition to Sanger sequencing, we used RNA sequencing data to call mutations from our PanNET samples specifically focusing on *ATRX/DAXX/MEN1* genes. We were able to confirm all mutations found by Sanger sequencing of DNA from the RNA transcripts of *ATRX/DAXX/MEN1* from RNA sequencing. However, we found no detectable SNVs (removing all SNPs from database dbSNP) and indels in coding region of *ATRX/DAXX/MEN1* from RNA sequencing in WT samples. Based on this mutational analysis from RNAseq, we believe there are no detectable SNV or indel mutations in coding region of *ATRX/DAXX/MEN1* in the WT samples.

Furthermore, we have performed ALT experiments and found strong correlation for ALT and *DAXX/ATRX* mutants PanNETs (74%) as compared to *DAXX/ATRX* WT PanNETs (12%). Since ALT is not exclusively present in *ATRX/DAXX* mutants nor do all *ATRX/DAXX* mutants have ALT, it cannot be used as a surrogate for *DAXX/ATRX* mutations⁵. Similarly, our data from immunohistochemistry demonstrate that while *DAXX* protein loss is identified in 91% PanNETs with *DAXX* mutation, only 60% *ATRX* protein expression loss is detected in *ATRX* gene mutated cases.

• *It is not clear why 33 samples used for the first round of experiments and only additional 14, instead for the rest of the them, for validation.*

Response:

We apologize for this confusion. We first performed gene expression microarray on 47 samples and then performed RNA sequencing on a random set of 33 of those 47 samples to obtain a more comprehensive transcriptome from sequencing. In our presentation, we began with analysis of the 33 RNA sequencing samples and then presented all 47 samples (33 original samples and 14 new samples) as confirmation using a different platform for gene expression (microarrays). We changed the word “validation” to “confirmation” in the revised manuscript (Page 5).

• *It is concerning that cell type specific markers do not show a pattern consistent with the hypothesis that A-D-M come from alpha cells. For example GCG expression, a marker for alpha cells (Murarno et al; Lawlor et al), is not specific for this group. Similar situation for INS, a beta cell, should not be expressed in this group.*

Response:

We believe the alpha-cell signature described in this study represents a set of genes which are responsible for the early differentiation of subtypes of pancreatic neuroendocrine cells during development; and they are distinct from genes corresponding to specific hormone production in mature neuroendocrine cells. The majority of our PanNETs are nonfunctional (26 of 33) which are probably not derived from differentiated pancreatic islet cells. They do not over produce or release neuroendocrine hormones leading to a specific clinical phenotype. Immunohistochemical studies demonstrate that nonfunctional PanNETs may express heterogenous peptide hormones of pancreatic islets or none of them at all; thus their corresponding homogeneous gene expression (GCG, INS etc.) is not expected. Indeed, some of the tumors in this study express combinations of neuroendocrine hormones such as GCG, INS, and SST. The heatmap below has been added to the supplemental figures along with the text in the discussion. To create robust gene signatures that are not sensitive to changes in expression of a few genes, we use a large number of genes to create the A-D-M mutant and alpha cell signatures.

Supplementary Figure 8:

Supplementary Figure 8: Heatmap of key pancreatic islet cell specific markers for different neuroendocrine cells and functional PanNETs hormone.

- It will be interesting to see the classification using supervised analysis on mutations.

Response:

We only sequenced *MEN1*, *ATRX*, and *DAXX* gene. As reported in our manuscript, *ATRX* and *DAXX* mutations were mutually exclusive but can co-occur with *MEN1* mutations. However, we could not detect significant gene expression differences between tumors with different A-D-M genotypes of which there were five: *MEN1* only, *DAXX* only, *ATRX* only, *MEN1* with *DAXX*, and *MEN1* with *ATRX* mutations. Instead, what we observe is that PanNETs with mutations in any combination of *ATRX*, *DAXX*, *MEN1* mutations have similar gene expression profile and were a more homogeneous group of PanNETs by gene expression than WT PanNETs.

Reviewer #2 (Remarks to the Author):

The most interesting part of the study is that the data of RNA sequencing and DNA methylation assay showed that A-D-M mutant PanNETs have a distinct gene expression and DNA methylation patterns from that of A-D-M WT PanNETs, suggesting the importance of ATRX, DAXX and MEN1 in regulating PanNETs. The authors, using the single cell sequencing study, also found that the gene expression profile of the A-D-M mutant PanNETs strongly correlates to the profile of the genes that are specifically expressed in alpha cells, including genes known to define alpha cells such as ARX. In general, understanding of the origin PanNETs, tumor progression and pathways is important as it may help to detect novel targets for PanNET treatment. This study has the potential to explain the unpredictable outcome of the PanNETs and facilitate the development of unique and targeted therapeutic strategies. However, some concerns are included as follows and must be addressed:

1. Figure 1b clearly showed that among 44 patients who initially presented with localized PanNETs (without distant metastasis), those with A-D-M mutant genotype had a worse recurrence free survival outcome than those with A-D-M WT genotype in their primary tumors. How about the survival outcome of the PanNETs with A-D-M mutant and metastasis?

Response:

The reviewer has raised a prudent clinical question. However, in this study we specially selected primary PanNETs to assess the probable metastatic potential; thus without enough cases of metastatic tumors, we cannot answer the question in this study set. Nevertheless, previously published studies have addressed this issue^{6,7}.

2. *HNFI1A*, *ARX*, *IRX2*, and *TM4SF4* were highly expressed in A-D-M mutant PanNETs compared to A-D-M WT PanNETs. However, did the authors perform the qRT-PCR or Western Blot to confirm that *ATRX*, *DAXX* and *MEN1* indeed regulate these genes using the PanNET cell lines? This would provide significant information regarding the functional importance of this correlation.

Response:

We thank the reviewer for this comment. We have not performed in vitro functional studies to show whether *ATRX*, *DAXX*, and *MEN1* regulate *HNFI1A*, *ARX*, *IRX2*, and *TM4SF4*. We are making a claim of association and not a causal claim of *MEN1/ATRX/DAXX* regulating *HNFI1A* etc. We found some evidence that *MEN1* may not directly regulate *HNFI1A* etc. Several studies have shown that loss of *MEN1* does not alter global H3K4me3 level but are responsible for altering H3K4me3 at specific loci⁸. We examined H3K4me3 ChIP-Seq and gene expression data available for *MEN1-WT* and *MEN1-null* mouse pancreatic islets, embryonic stem cells and pancreatic islet-like endocrine cells⁸, and we did not find differential expression of *HNFI1A*, *ARX*, *IRX2*, and *TM4SF4* or altered H3K4me3 at their promoters in *MEN1-WT* and *MEN1-null* mice.

3. The authors did DNA methylation analysis and found that DNA methylation is distinct between the A-D-M mutant and A-D-M WT PanNETs. It is well documented that menin is very crucial for epigenetic regulation by histone modifications, such as H3K4me3 and H3K9me3. Therefore, whether the histone modifications are different between the A-D-M mutant and A-D-M WT PanNETs and whether the histone modifications contribute to the change of gene expression and DNA methylation? A functional and mechanistic study will surely strengthen the manuscript.

Response:

We completely agree with you that additional functional and mechanistic studies regarding biochemical studies of histone modifications and their interplay with DNA methylation and gene expression will be

interesting to pursue but we feel this is beyond the scope of this paper; and we trust that investigators with expertise in functional cell biopsy and biochemistry will carry out the study in the near future.

4. High ARX gene expression was observed with the promoter and first exon of ARX not differentially methylated, while low PDX1 gene expression was observed with PDX1 promoter hyper-methylation in the A-D-M mutant PanNETs as compared to A-D-M WT PanNETs. However, do ATRX, DAXX and MEN1 promote or repress the DNA methylation? Which is caused by ATRX, DAXX and MEN1 mutation? Certain cellular and functional studies would much improve the manuscript.

Response:

We did not observe global increase or decrease in the DNA methylation level in A-D-M mutant compared to A-D-M WT panNETs (Result section part 6, Page 5). What we observed is that some DNA methylation sites were different between the A-D-M mutant and WT panNETs. We agree that pursuing cellular and functional studies regarding how ATRX, DAXX, and MEN1 may alter DNA methylation would be interesting to pursue but we feel this is beyond the scope of this paper.

5. The frequencies of ATRX, DAXX and MEN1 mutations are 8%, 16% and 20%, respectively. Are these mutations found previously? Are these mutations reported to correlate to alternation of DNA methylation? A discussion about this would be of help.

Response:

Thank you for the suggestion. We regret the error. We had initially included the mutation information in a supplemental file but it did not appear. The mutation information was part of clinical supplementary file 1 as an excel file and found in sheet 2. However, there was a problem in file conversion from excel to pdf, so sheet 2 was missing. We have now created a separate file for mutation information.

Mutations in *ATRX* and *DAXX* were all unique. For *MEN1*, all mutations were unique with the exception of one in-frame (deletion) mutation that was identical for three samples (this mutation is a germline mutation with LOH in the tumor and the three individuals are related). The majority of the mutations in *ATRX*, *DAXX*, *MEN1* were truncation mutations (stopgain or frameshift) and loss of function, consistent with their role as tumor suppressors in panNETs. We have added this information to the paper in result section part 1 (Page 3). “The majority of mutations in *ATRX*, *DAXX*, *MEN1* were truncation mutations (stopgain or frameshift) and loss of function consistent with their role as tumor suppressors (Supplemental file 1b).”

We did not observe differences in DNA methylation pattern between truncation and missense mutations.

Minor:

1. Lane 47, 222, 223...: *MEN1* is the name of multiple endocrine neoplasia type1 and should be replaced by italic *MEN1* or *menin*, the protein name of *MEN1*.

Response: Thank you for pointing this out. We have made appropriate changes in the revised manuscript.

2. Lane 145, two “that”.

Response: We have removed one “that” at that line.

3. Lane 240-241: “Alpha cell-specific *Men1* ablation triggers the transdifferentiation of glucagon-expressing cells and insulinoma development” from Zhang CX group also found that deletion of *MEN1* in alpha cells developed insulinomas.

Response: We appreciate the provided reference and it has been added to the revised manuscript.

4. “Conditional deletion of *Men1* in the pancreatic β -cell leads to glucagon-expressing tumor development” from Guang Ning group found that specific *Men1* knockout in β -cell developed both glucagon-expressing tumors and insulinomas. These glucagon-expressing tumor cells were *menin* deficient and expressed the mature α -cell-specific transcription factors Brain-specific homeobox POU domain protein 4 (*Brn4*) and *v-maf* musculoaponeurotic fibrosarcoma oncogene family, protein B (*MafB*). This reference should be included.

Response: We appreciate the provided reference and it has been added to the revised manuscript.

Reviewer #3 (Remarks to the Author):

This manuscript described a subset of pancreatic neuroendocrine tumors by three gene mutations, supported by gene expression and DNA methylation profiles, as a distinct alpha-cell subgroup, which appears having a clinical implication (i.e., patient survival). Although valuable to the medical/research community, the writing and analyses are a bit shallow. Lack of an independent validation is another concern for its certainty. Below are some specific points:

1) *The background did not provide sufficient information in the topic. What other mutations are commonly found in this tumor? What are known predictors for clinical outcomes or classification methods for patient management (the authors mentioned grade and stage but did not say their clinical relevance)?*

Response: The reviewer’s point is well taken, and we have provided the relevant information in the Introduction section of the revised manuscript (Page 2).

2) *The method or result did not provide sufficient details about the mutations in these three genes. Do patients have the exact same mutation across individuals for the same category of mutation (missense, in-frame or truncating...)? For example, in-frame mutations are most common in this set of patients but are they exactly the same (at the same location, same length, insertion or deletion)? The supplementary data seems not including the detailed mutation information (or not uploaded).*

Response:

Thank you for the suggestion. We regret the oversight. We had initially included the mutation information in a supplemental file but it did not appear due to a problem encountered during file conversion from excel to PDF. The mutation information was part of clinical supplementary file 1 as an excel file and found in sheet 2. We have now created a separate file for mutation information.

Mutations in *ATRX* and *DAXX* were all unique. For *MEN1*, all mutations were unique with the exception of one in-frame (deletion) mutation that was identical for three samples (this mutation is a germline mutation with LOH in the tumor and the three individuals are related). The majority of the mutations in *ATRX*, *DAXX*, *MEN1* were truncation mutations (stopgain or frameshift) and loss of function, consistent with their role as tumor suppressors in panNETs. We have added this information to the paper in result section part 1 (Page 3). “The majority of mutations in *ATRX*, *DAXX*, *MEN1* were truncation mutations (stopgain or frameshift) and loss of function consistent with their role as tumor suppressors (Supplemental file 1b).”

3) *In addition to the survival, authors should also correlate to other clinical variables such as grade and stage. Multivariable survival analysis is also needed to see if the classification by the genotype makes difference.*

Response:

We thank the reviewer for raising this clinically relevant comment. The cases of PanNETs in this study were randomly selected from a larger cohort (Memorial Sloan-Kettering Cancer Center) which have been studied extensively with regard to the clinicopathologic correlations in our previously published work⁹. Nevertheless, we did perform statistic analysis on the current data set separately and this information has been included in the revised manuscript (Page 3). Due to limitation for number of figures, we cannot including additional figures in this manuscript.

4) *The authors should conduct comparisons with public data such as the two cited (ref 2,3). In Jiao’s paper, they found the tumors with mutations in these three genes were associated with better survival but here it is reverse so further exploration and explanation are needed.*

Response:

We appreciate the reviewer's suggestion. Unfortunately, the comparison of data sets and clinical outcome between studies across different institutions and different continents with different analytical platforms would be difficult and challenging. In the initial study by Jiao et al.,⁴ the specimens included liver metastases from PanNETs only and no primary tumors were included; in contrast, the current study focus on primary PanNETs. In the study by Scarpa et al.,¹ the specimens were collected from Verona Italy and multiple institutions in Australia, and no clinical data regarding prognosis or survival were provided. With the exception of the initial study by Jiao et al.,⁴ a number of more recent studies have shown reciprocal finding of adverse prognosis of PanNETs with DAXX/ATRX mutations^{6,7}. We have included additional discussion of this point in the revised manuscript (Page 8).

5) *The integrative analysis between methylation and gene expression is weak. What is the set of genes that are likely impacted by DNA methylation and what is the set of genes that are not affected (in the mutation context)? Is the mutant group more hyper or hypo-methylated? Where are the changes mostly located in the genome (CpG Island, shore, shelf, intergenic region....)? Do the mutations affect DNA methylation? I guess the main question is whether the association is causal or coincidence.*

Response:

We found 59 genes, which are differentially methylated at TSS1500/200 and first exon between A-D-M mutants and A-D-M WT panNETs. Thirteen of the 59 genes were also differentially expressed.

- Eight of them are likely impacted by DNA methylation. Seven genes hypomethylated in A-D-M mutant and over-expressed are *APOH*, *CCL15*, *EMID2*, *PDZK1*, *HAO1*, *BAIAP2L2*, and *NPC1L1*. One gene, *TACR3*, was hypomethylated in A-D-M WT and over-expressed.

- Three genes, *DLG2*, *MMP26* and *GJA3* have DNA methylation status at TSS1500/200 and first exon that is not consistent with gene expression. *MMP26* and *GJA3* are over expressed and hypermethylated in A-D-M mutants and *DLG2* is over expressed and hypermethylated in WT panNETs.

- Two gene, *C20orf118* and *F2* are overexpressed in A-D-M mutants and have DNA hemimethylation status (beta value ≥ 0.3 and ≤ 0.7).

We did not observed global hypo or hypermethylation in A-D-M mutant or WT panNETs.

We found 51% of differentially methylated probes at opensea, 12% at CpG Island, 10% at Shelf and 27% at shore part of the genome (based on 450K chip array).

We changed the last results section title to “Integrative analysis reveal PDX1 gene is hypermethylated with low expression in A-D-M mutant PanNETs” and included additional information (Page 7-8).

We do not know if the difference in DNA methylation between A-D-M mutant and WT panNETs are due to mutations in ATRX, DAXX, and MEN1. What we found is an association between DNA methylation and gene expression with mutation status. To explore whether there is a causal effect is an interesting area to pursue requiring additional functional studies, which we feel is beyond the scope of this paper.

References for responses:

- 1 Scarpa, A. *et al.* Whole-genome landscape of pancreatic neuroendocrine tumours. *Nature* **543**, 65-71, doi:10.1038/nature21063 (2017).
- 2 Sadanandam, A. *et al.* A Cross-Species Analysis in Pancreatic Neuroendocrine Tumors Reveals Molecular Subtypes with Distinctive Clinical, Metastatic, Developmental, and Metabolic Characteristics. *Cancer Discov* **5**, 1296-1313, doi:10.1158/2159-8290.CD-15-0068 (2015).
- 3 Yoshihara, K. *et al.* Inferring tumour purity and stromal and immune cell admixture from expression data. *Nat Commun* **4**, 2612, doi:10.1038/ncomms3612 (2013).
- 4 Jiao, Y. *et al.* DAXX/ATRX, MEN1, and mTOR pathway genes are frequently altered in pancreatic neuroendocrine tumors. *Science* **331**, 1199-1203, doi:10.1126/science.1200609 (2011).
- 5 Heaphy, C. M. *et al.* Altered telomeres in tumors with ATRX and DAXX mutations. *Science* **333**, 425, doi:10.1126/science.1207313 (2011).
- 6 Marinoni, I. *et al.* Loss of DAXX and ATRX are associated with chromosome instability and reduced survival of patients with pancreatic neuroendocrine tumors. *Gastroenterology* **146**, 453-460 e455, doi:10.1053/j.gastro.2013.10.020 (2014).
- 7 Park, J. K. *et al.* DAXX/ATRX and MEN1 genes are strong prognostic markers in pancreatic neuroendocrine tumors. *Oncotarget* **8**, 49796-49806, doi:10.18632/oncotarget.17964 (2017).
- 8 Lin, W. *et al.* Dynamic epigenetic regulation by menin during pancreatic islet tumor formation. *Mol Cancer Res* **13**, 689-698, doi:10.1158/1541-7786.MCR-14-0457 (2015).
- 9 Ferrone, C. R. *et al.* Determining prognosis in patients with pancreatic endocrine neoplasms: can the WHO classification system be simplified? *J Clin Oncol* **25**, 5609-5615, doi:10.1200/JCO.2007.12.9809 (2007).

Reviewers' comments:

Reviewer #1 (Remarks to the Author):

Several studies have demonstrated that a large portion of PanNETs have mutation in MEN1 and/or ATRX/DAXX, but a large proportion does not have mutations in these genes. The primary PanNETs without mutations in these genes still metastasize but overall they have better prognosis. It has been hypothesized that the cell of origin is different between the PanNETs with mutation in these genes, called here A-D-M and WT. This study suggests that the cell of origin for A-D-Ms is the alpha pancreatic cell based on expression profiles.

Comments:

- This study provides preliminary data that the alpha type is the cell of origin for PanNETs with mutations in MEN1 and/or ATRX/DAXX. However, there is not any classifier presented here that can be used for this purpose and the expression profile is not validated well in an independent set of samples.
- The text indicates that tumor purity is 80%, but this is a pathological estimate. It will be good to see percent mutant in the samples that have mutations. It will be a more accurate way to determine purity. The upregulation of liver specific genes is curious and contamination needs to be excluded the best way possible.
- The fraction of the samples with DAXX or ATRX mutations appear lower than what has been previously published. Depending on the method for mutation detection, some mutations will be missed. Have the samples negative for mutations been checked for ALT, as a surrogate for ATRX/DAXX mutations, or for the absence/presence of ATRX or DAXX with an antibody? This will ensure that the WT group is indeed WT.
- It is not clear why 33 samples used for the first round of experiments and only additional 14, instead for the rest of them, for validation.
- It is concerning that cell type specific markers do not show a pattern consistent with the hypothesis that A-D-M come from alpha cells. For example GCG expression, a marker for alpha cells (Murano et al; Lawlor et al), is not specific for this group. Similar situation for INS, a beta cell marker, should not be expressed in this group.
- It will be interesting to see the classification using supervised analysis on mutations.

Reviewer #2 (Remarks to the Author):

The most interesting part of the study is that the data of RNA sequencing and DNA methylation assay showed that A-D-M mutant PanNETs have a distinct gene expression and DNA methylation patterns from that of A-D-M WT PanNETs, suggesting the importance of ATRX, DAXX and MEN1 in regulating PanNETs. The authors, using the single cell sequencing study, also found that the gene expression profile of the A-D-M mutant PanNETs strongly correlates to the profile of the genes that are specifically expressed in alpha cells, including genes known to define alpha cells such as ARX. In general, understanding of the origin PanNETs, tumor progression and pathways is important as it may help to detect novel targets for PanNET treatment. This study has the potential to explain the unpredictable outcome of the PanNETs and facilitate the development of unique and targeted therapeutic strategies. However, some concerns are included as follows and must be addressed:

1. Figure 1b clearly showed that among 44 patients who initially presented with localized PanNETs (without distant metastasis), those with A-D-M mutant genotype had a worse recurrence free survival outcome than those with A-D-M WT genotype in their primary tumors. How about the survival outcome of the PanNETs with A-D-M mutant and metastasis?
2. HNF1A, ARX, IRX2, and TM4SF4 were highly expressed in A-D-M mutant PanNETs compared to A-D-M WT PanNETs. However, did the authors perform the qRT-PCR or Western Blot to confirm that ATRX, DAXX and MEN1 indeed regulate these genes using the PanNET cell lines? This would

provide significant information regarding the functional importance of this correlation.

3. The authors did DNA methylation analysis and found that DNA methylation is distinct between the A-D-M mutant and A-D-M WT PanNETs. It is well documented that menin is very crucial for epigenetic regulation by histone modifications, such as H3K4me3 and H3K9me3. Therefore, whether the histone modifications are different between the A-D-M mutant and A-D-M WT PanNETs and whether the histone modifications contribute to the change of gene expression and DNA methylation? A functional and mechanistic study will surely strengthen the manuscript.

4. High ARX gene expression was observed with the promoter and first exon of ARX not differentially methylated, while low PDX1 gene expression was observed with PDX1 promoter hyper-methylation in the A-D-M mutant PanNETs as compared to A-D-M WT PanNETs. However, do ATRX, DAXX and MEN1 promote or repress the DNA methylation? Which is caused by ATRX, DAXX and MEN1 mutation? Certain cellular and functional studies would much improve the manuscript.

5. The frequencies of ATRX, DAXX and MEN1 mutations are 8%, 16% and 20%, respectively. Are these mutations found previously? Are these mutations reported to correlate to alternation of DNA methylation? A discussion about this would be of help.

Minor:

1. Lane 47, 222, 223...: MEN1 is the name of multiple endocrine neoplasia type1 and should be replaced by italic MEN1 or menin, the protein name of MEN1.

2. Lane 145, two "that".

3. Lane 240-241: "Alpha cell-specific Men1 ablation triggers the transdifferentiation of glucagon-expressing cells and insulinoma development" from Zhang CX group also found that deletion of MEN1 in alpha cells developed insulinomas.

4. "Conditional deletion of Men1 in the pancreatic β -cell leads to glucagon-expressing tumor development" from Guang Ning group found that specific Men1 knockout in β -cell developed both glucagon-expressing tumors and insulinomas. These glucagon-expressing tumor cells were menin deficient and expressed the mature α -cell-specific transcription factors Brain-specific homeobox POU domain protein 4 (Brn4) and v-maf musculoaponeurotic fibrosarcoma oncogene family, protein B (MafB). This reference should be included.

Reviewer #3 (Remarks to the Author):

This manuscript described a subset of pancreatic neuroendocrine tumors by three gene mutations, supported by gene expression and DNA methylation profiles, as a distinct alpha-cell subgroup, which appears having a clinical implication (i.e., patient survival). Although valuable to the medical/research community, the writing and analyses are a bit shallow. Lack of an independent validation is another concern for its certainty. Below are some specific points:

1) The background did not provide sufficient information in the topic. What other mutations are commonly found in this tumor? What are known predictors for clinical outcomes or classification methods for patient management (the authors mentioned grade and stage but did not say their clinical relevance)?

2) The method or result did not provide sufficient details about the mutations in these three genes. Do patients have the exact same mutation across individuals for the same category of mutation (missense, in-frame or truncating...)? For example, in-frame mutations are most common in this set of patients but are they exactly the same (at the same location, same length, insertion or deletion)? The supplementary data seems not including the detailed mutation information (or not uploaded).

3) In addition to the survival, authors should also correlate to other clinical variables such as grade and stage. Multivariable survival analysis is also needed to see if the classification by the genotype makes difference.

4) The authors should conduct comparisons with public data such as the two cited (ref 2,3). In Jiao's paper, they found the tumors with mutations in these three genes were associated with better survival but here it is reverse so further exploration and explanation are needed.

5) The integrative analysis between methylation and gene expression is weak. What is the set of genes that are likely impacted by DNA methylation and what is the set of genes that are not affected (in the mutation context)? Is the mutant group more hyper or hypo-methylated? Where are the changes mostly located in the genome (CpG Island, shore, shelf, intergenic region...)? Do the mutations affect DNA methylation? I guess the main question is whether the association is causal or coincidence.

REVIEWERS' COMMENTS:

Reviewer #1 (Remarks to the Author):

The authors addressed my comments; however, the data do not strongly support the finding. It is also not clear how useful the signature that defines ADM is and how is going to be used in the future, both clinically and scientifically.

I remain skeptical, that the data do not appear as “clean” as they could be due to misclassification of some of the samples. For example, in their response the authors claim that 12% of ADM WT have ALT. Although there are examples in the literature where a mutation in ATRX/DAXX has not been identified in ALT cases (because it has been missed or because another gene in the pathway is mutated), the opposite is not true. That is that all ATRX/DAXX oncogenic mutations result in ALT. So, the 74%, 12% split could actually result in “contamination” of the subgroups resulting in less robust results.

I still think that the best profile could be generated by dividing the samples in ATRX/DAXX/ALT and WT, and then look at the combination with MEN1 mutations. I do understand that this requires more samples which may not be available, at least, the conclusions need to be toned down and these issues need to be discussed.

I think that the new supplementary figure 8 confuses things more.

Response to Reviewer 1

We thank reviewer #1 for pointing out the discrepancy where only 74% of the ATRX/DAXX mutant PanNETs were found to have the ALT phenotype while 12% of the ADM WT tumors did have ALT. We had initially used the Telomere Restriction Fragment (TRF) assay for those results. We have subsequently re-assayed for ALT using FISH and found that 93% of ATRX/DAXX mutant tumors have ALT while 12% of ADM WT tumors have ALT. In fact, the only ATRX/DAXX mutant tumor that was negative for ALT by FISH assay was positive for ALT by TRF assay. This new result is now in agreement with the reviewer's important point that all ATRX/DAXX mutant PanNETs would be expected to exhibit ALT. In light of the new ALT results, we believe we have addressed all the concerns from the reviewer regarding “contamination”/misclassification of the PanNETs. With the new ALT results where all ATRX/DAXX mutant tumors are ALT positive, the reviewer's suggestion of generating a profile by dividing the samples into ATRX/DAXX/ALT and ADM WT produces a gene expression profile for ATRX/DAXX/ALT that is nearly identical to the ADM mutant profile. Therefore, we believe our data strongly support our findings.

We used gene expression to classify the PanNETs because in an initial unsupervised clustering analysis (without supervised selection based on genotype or ALT status), gene expression naturally produced two very distinct groups of PanNETs that is robust to number of genes used for clustering. A surprising and significant finding of this work was that one of the two clusters of PanNETs is comprised almost exclusively of tumors with ADM mutations. Moreover, the

ADM gene signature is sufficiently similar to the alpha cell signature to suggest that alpha cells or alpha-like cells may be the cells of origin of PanNETs with ADM mutations. To study the epigenetic and transcriptional dysregulation occurring in ADM PanNETs, the actual cells of origin must be used for comparison. Thus, this study represents a novel observation with considerable scientific significance since beta cells have generally been considered to be the cell of origin of PanNETs. The findings have provided insight to direct future studies of ADM PanNETs, which has the potential to identify diagnostic, prognostic, and therapeutic strategies of clinical significance for this previously less-well-understood disease.

We agree with the reviewer that the new supplementary figure 8 is confusing and have removed it.

Reviewer #2 (Remarks to the Author):

Overall, the authors did a good job in addressing the questions and improving the manuscript.

It is not immediately clear whether the authors have submitted the detailed GSE files for both the microarray data and RNA seq data from the PNET samples for deposit. It is mandatory to submit these data sets to freely disseminate the data and also to allow the comparison of the current data with the previously published data.

With all the data sets are submitted and deposited for free access by scientists, this reviewer thinks that the MS is ready for publication.

Response to Reviewer 2

We have submitted all microarray data, 450K methylation data and RNA seq data to GEO with accession number GSE117853 and have inserted the reference in the manuscript.

Reviewer #3 (Remarks to the Author):

The authors have addressed the concerns adequately and I have no further comments. I appreciate the authors' efforts.

Response to Reviewer 3

We thank the reviewer's positive comment and for taking the time and effort to review our manuscript.